# Potential Therapeutic Effect of All-Trans Retinoic Acid on Atherosclerosis

**DOI:** 10.3390/biom12070869

**Published:** 2022-06-22

**Authors:** Qile Deng, Jixiang Chen

**Affiliations:** Department of Neurology, Union Hospital, Tongji Medical College, Huazhong University of Science and Technology, Wuhan 430022, China; m201975864@hust.edu.cn

**Keywords:** ATRA, atherosclerosis, adipocytes, RARs, immunity

## Abstract

Atherosclerosis is a major risk factor for myocardial infarction and ischemic stroke, which are the leading cause of death worldwide. All-trans retinoic acid (ATRA) is a natural derivative of essential vitamin A. Numerous studies have shown that ATRA plays an important role in cell proliferation, cell apoptosis, cell differentiation, and embryonic development. All-trans retinoic acid (ATRA) is a ligand of retinoic acid receptors that regulates various biological processes by activating retinoic acid signals. In this paper, the metabolic processes of ATRA were reviewed, with emphasis on the effects of ATRA on inflammatory cells involved in the process of atherosclerosis.

## 1. Introduction

Atherosclerosis is a progressive disease of large arteries and a leading cause of stroke and cardiovascular disease [1]. All in all, atherosclerosis can be considered to be a chronic inflammation caused by the interaction between monocyte-derived macrophages, T cells, and other immune response cells [2]. When hypertension, hypercholesterolemia, and other persistent effects or blood flow shear forces from vascular bifurcation diseases cause endothelial cell dysfunction or anatomical damage, vascular smooth muscle cells and fibroblast migrate and proliferate to the intimal layer wall [2]. This inflammatory process can eventually result in complex lesions or the development of plaques that can narrow blood vessels and interfere with normal blood flow to the heart and brain, leading to clinical symptoms [3]. According to statistics, the number of deaths caused by cardiovascular and cerebrovascular diseases accounts for nearly one-third of all the deaths in the world, and this number will increase year by year [4]. Although the common risk factors for atherosclerosis such as hyperlipidemia, hypertension, hyperglycemia, and smoking have been strictly controlled, the incidence of atherosclerotic diseases remains high. Thus, developing new strategies for prevention and treatment requires understanding other factors contributing to atherogenesis.

Vitamin A participates in a variety of physiological processes, such as embryonic development, energy metabolism, immune regulation, and other functions [5]. ATRA plays an important role in the cell growth, development, and differentiation of vertebrates [6]. ATRA has been implicated in several diseases, including inflammatory disorders [7] and cancer [8]. Currently, ATRA is the routine treatment for the management of acute promyelocytic leukemia (APL) [9]. Studies have reported that ATRA can significantly inhibit the formation of atherosclerotic lesions in rabbit models of atherosclerosis induced by a high-fat diet [10,11]. In addition, it was shown that ATRA can inhibit restenosis after balloon angioplasty in atherosclerotic rabbits [12]. A recent study showed that ATRA relieves coronary artery stenosis by regulating the function of smooth muscle cells in a mouse model of Kawasaki disease [13].

This article reviews the effects of ATRA and retinoic acid (RA) on the development of atherosclerotic cardiovascular disease (ASCVD). First, we focused on ATRA molecular signal transduction, and then we discussed the effects of ATRA on lipid metabolism and various inflammatory cells. The main focus of this review is the effect of ATRA on various types of inflammatory cells, and then it will discuss a summary of the current literature related to the physiology of ATRA and macrophages, T cells, and smooth muscle cells.

## 2. Metabolism of ATRA

Vitamin A achieves its effect on growth and development through the action of the biologically active metabolite RA, mainly including ATRA and 9-cis-retinoic acid (9-cis-retinoic acid) [14,15]. However, all-trans-RA is the primary ligand during development [16]. Mammals cannot synthesize vitamin A, they can only get vitamin A from food. Vitamin A mainly exists in the form of retinol ester, which is hydrolyzed to retinol by retinol ester hydrolase (REH). Retinol binds to the retinol binding protein (RBP4) in the plasma and enters the cell, where it is oxidized by retinol dehydrogenase (RDH) to all-trans retinal, and then by retinol dehydrogenase (ALDH) converted to ATRA (Figure 1). Cytochrome P450 superfamily enzymes (CYP26A1, CYP26B1, and CYP26C1) strictly regulate the intracellular concentration of ATRA and promote the conversion of intracellular ATRA into inactive metabolites [17]. The effects of ATRA were relayed by binding to nuclear retinoid receptors [18]. Two types of nuclear retinoic acid receptors (RARs) and retinoic X receptors (RXRs) have been discovered [19,20]. ATRA only binds to retinoic acid receptors (RARs) with high affinity, 9-cis retinoic acid is an isomer of ATRA which can be combined with RARs and retinoic X receptors (RXRs) [21]. 13-cis retinoic acid, another isomer of ATRA, is a high-affinity ligand that only binds to RXRs [22]. Among RA isomers, ATRA can bind to and activate RARs both in vivo and in vitro. 9-cis-RA can activate RXRs and RARs in vitro, but it does not work in the body [23]. 13-cis-RA makes up about 25% of circulating retinoic acid levels [24]. 9-cis RA has been detected at extremely low levels. Therefore, most of the in vivo activation of all RAR subtypes (α, β and γ) is considered to be ATRA-mediated [23].

RARs are members of the steroid/vitamin D/thyroid hormone receptor family, the pleiotropic effect of ATRA is mediated through its combination with RARs [18], and regulates many biological functions, such as embryonic development, organogenesis, homeostasis, vision, immune function, and reproduction [25]. Like most nuclear receptors (NRs), RARs have the function of the dimer. RARs cooperate with the retinoid receptor (RXR) to form heterodimers, which are the main regulators of human gene expression and an important drug target [26]. RARs and RXRs are modular proteins composed of several domains, the most notable being a DNA binding domain (DBD) and a C-terminal ligand-binding domain (LBD) [27]. The LBD contains the ligand-dependent activation function, AF-2 [27]. Before binding to the ligand, the RXR and RARα bind to specific regions of the retinoic acid response element (RARE) DNA as heterodimers. The RXR and RARα recruit silencing mediators of retinoic acid and thyroid hormone receptor (SMRT), nuclear receptor corepressor (NCoR), a corepressor related to histone deacetylase (HDAC), leading to transcriptional inhibition of target genes [28]. After binding to ATRA, the conformation of the RARα changes, triggering the release of co-inhibitors and the recruitment of co-activators (steroid receptor co-activator, SRC-1, -2, and -3) and the recruitment of histone acetyltransferases (HATs), leading to transcriptional activation of target genes [29] (Figure 2). ATRA binds to retinoid receptors and their heterodimers, which can affect cellular processes not only through genomic pathways but also through non-genomic mechanisms [30].

## 3. ATRA and Adipocytes

Adipose tissue accounts for 15 to 20 percent of the body’s total steroid reserves [31] and is a potential target for the effects of ATRA [32]. The differentiation of adipose precursor cells into adipose cells in white (WAT) or brown adipose tissue (BAT) is related to the accumulation of intracellular retinoids [33]. In addition to storing retinoids, adipocytes also synthesize and secrete RBP [34]. Vitamin A is a nutrient that has significant effects on adipose tissue biology and energy homeostasis [35]. Dietary vitamin A and provitamin A are stored in the form of retinol, or metabolized in cells to retinoic acid, the main active form of vitamin A [36]. All-trans retinoic acid treatment can reduce fat in mice and improve insulin sensitivity by promoting fat mobilization and catabolism [37]. In addition, ATRA can effectively inhibit the differentiation of clonal preadipocyte lines in vitro [38]. ATRA reduces the ability of adipogenesis, directly increases lipolysis, and reduces the triacylglycerol content in mature adipocytes [39]. Tourniaire et al. [40] showed that ATRA affects mitochondria in adipocytes, resulting in increased oxidative phosphorylation (OXPHOS) capacity and mitochondrial content in these cells. Exogenous ATRA acts mainly by reducing the proliferation and differentiation of preadipocytes, inducing apoptosis, and promoting adipocyte defatting [41]. Perivascular adipose tissue (PVAT) is composed of a mixture of two types of WAT and brown adipose tissue (BAT) [42]. Apolipoprotein E-deficient mice (Apo-E) treated with ATRA stimulated PVAT browning and increased adiponectin synthesis. Apo-E mice treated with ATRA stimulated PVAT browning and increased adiponectin synthesis while improving atherosclerosis in Apo-E mice [43]. However, further studies are needed to demonstrate the role of ATRA in regulating lipid metabolism in vivo. The molecular mechanism of ATRA’s influence on lipid metabolism is complex and not yet fully understood.

## 4. Effect of All-Trans Retinoic Acid on Atherosclerosis

The pathological basis of most cardiovascular and cerebrovascular diseases is atherosclerosis, which is the main cause of death worldwide. Atherosclerosis is a chronic disease involving processes such as lipid deposition, endothelial damage, and immune inflammation. Endothelial cells, macrophages, white blood cells, and intimal smooth muscle cells are the main players in the development of this disease [44]. So far, atherosclerosis is still the most common underlying cause of cardiovascular and cerebrovascular diseases. ATRA affects the development of atherosclerosis by regulating the biological processes of immune cells, such as cell proliferation, migration, and phenotypic transformation. ATRA is also involved in the regulation of blood glucose concentration, lipid metabolism, inflammation, and other risk factors for atherosclerosis [45].

### 4.1. ATRA and Macrophages

In Western society, atherosclerotic cardiovascular disease is the main cause of death. The hallmark of early-stage atherosclerosis is the accumulation of macrophage-derived foam cells [46,47]. Due to the uncontrolled uptake of modified low-density lipoprotein (LDL) by macrophages, excessive lipoprotein-derived cholesterol accumulates in the cells, leading to the formation of macrophage foam cells. The removal of cholesterol deposits in arteries by macrophages is beneficial in the early stages of atherosclerosis [48]. HDL (high-density lipoprotein) particles enhance the outflow of cholesterol from foam cells, which is the first step of reverse cholesterol transport (RCT) [49]. The endogenous production of lipid-poor Apo-E and the ATP-binding cassette transporters A1 (ABCA1) and G1 (ABCG1) are common pathways for cholesterol efflux [50,51]. Ayaori et al. [52] found that ATRA enhanced the expression of ABCA1 and ABCG1 in ApoA-I/HDL-mediated THP-1 macrophages and human monocyte-derived macrophages (HMDM), thereby promoting cholesterol efflux from macrophages. The liver X receptor (liver X receptor, LXR) is a ligand-dependent nuclear receptor, which plays an important role in the cells involved in the transcriptional regulation of cholesterol homeostasis. ABCA1 is one of the most well-known LXR target genes, and LXR activation increases ABCA1 expression, which in turn promotes the outflow of cholesterol and phospholipids to lipid-poor ApoA-I, resulting in the formation of nascent HDL particles. The transactivation of RXRs can enhance the expression of LXR target genes, thereby regulating lipid homeostasis, which is an attractive therapeutic target in macrophages. The coordinated tissue-specific role of the LXR pathway maintains cholesterol homeostasis throughout the body and regulates immune and inflammatory responses [53]. Therefore, LXR agonists have attracted widespread attention in the treatment of atherosclerosis. Unfortunately, due to its serious adverse reactions, this class of drugs has not yet been approved by the FDA. Fortunately, the use of nanoparticles (NPs) to target the delivery of LXR agonists can facilitate effective LXR-mediated therapy [54].

Macrophages are plastic cells with proinflammatory or anti-inflammatory properties. In response to microenvironmental signals, two classical modes of macrophages are activated, proinflammatory M1 macrophages or anti-inflammatory M2 macrophages [55]. During the progression of atherosclerosis, M1 macrophages have been shown to aggravate plaque and systemic inflammation, leading to plaque rupture, which dominates the progression of atherosclerosis. M2-like macrophages are abundant in stable plaques [56]. ATRA has been found to promote M1 and M2 phenotypic transformation in tumors [57], parasitic infections [58], and other diseases, but no studies have shown that ATRA can promote the phenotypic transformation of macrophages in atherosclerosis. However, recent studies have shown that ATRA can enhance the initiating signal of NOD-like receptor protein-3 (NLRP3) inflammasome in human macrophages, thereby improving the inflammatory response [59].

### 4.2. ATRA and SMC

Smooth muscle cells (SMCs) are the most important cells involved in the development of atherosclerosis, and they tend to occur in the thickened intimal layer of arteries. With the development of atherosclerotic plaque and intimal SMC phenotype changes, contractility is lost, SMC marker expression decreases, and it participates in the formation of foam cells [60]. The phenotype of SMC is converted to a poorly differentiated form and SMC markers are reduced or completely absent, leading to an underestimation of its role in the development of atherosclerosis in humans and animals. A large number of studies have shown that ATRA is involved in the active process of vascular injury sites, including SMC differentiation and proliferation. It is a potential candidate to prevent atherosclerosis, intimal hyperplasia, and restenosis.

#### 4.2.1. ATRA and SMC Proliferation and Migration

The accumulation of smooth muscle cells (SMCs) in the intima is a characteristic of vascular lesions, including atherosclerotic plaque and restenosis after angioplasty. In the past few decades, there have been conflicting reports about the effects of retinoids on the proliferation of vascular smooth muscle cells. At the earliest, Peclo showed that ATRA has a mitogenic effect on smooth muscle cells [61]. Other studies have not found the effect of retinoic acid on SMC proliferation [62]. However, the vast majority of reports have reported the inhibitory effects of ATRA on the growth of vascular smooth muscle cells (VSMCs) of different species from rats to humans [63]. A number of studies have shown that several mitotic factors can inhibit the proliferation of SMCs, such as PDGF-BB (platelet-derived growth factor-BB) [63], endothelin-1, serum [64], serotonin [65], AngII (angiotensin II) [66], endothelin-1 [67], and bFGF (basic fibroblast growth factor) [68]. Krüppel-like zinc finger transcription factor 5 (KLF5; also known as BTEB2 and IKLF) was significantly induced in activated vascular smooth muscle cells and fibroblasts, whereas the arterial wall thickening of heterozygous KLF knockout (KLF5 +/−) mice was significantly lower than that of wild-type animals [69]. Shindo et al. [69] discovered that KLF5 interacts with the retinoic acid receptor (RAR), and the synthesis of RAR ligands regulates the transcriptional activity of KLF5 to play a protective effect on cardiovascular remodeling. In addition, Sakamoto and colleagues [70] showed that the growth of VSMCs in cultured human atherosclerotic specimens was associated with increased KLF5 expression. These studies clearly demonstrated the inhibitory effect of retinoic acid on the proliferation of SMC. Tran-Lundmark [71] demonstrated that ATRA inhibits the proliferation of SMCs by regulating the expression of perlecan in SMCs, depending on its heparin sulfate chain. In addition to its effect on perlecan, ATRA can also induce the expression of tumor suppressor a-kinase anchoring protein 12 in SMCs and block the growth of vascular SMCs [72], and has been shown to inhibit the expression of fibrin, an extracellular matrix component of human fetal palatal mesenchymal cells [73]. In SMCs, ATRA can downregulate urokinase plasminogen activator (uPA) [74]. uPA can activate a number of proteins, one of which is PDGF-D, an effective activator of SMC proliferation [75]. Recent studies on the response of VSMCs to vascular injury have shown that ATRA can inhibit intimal hyperplasia and smooth muscle cell proliferation and migration by activating the AMPK signaling pathway and inhibiting mTOR signaling [76].

In summary, retinoic acid can inhibit the proliferation and migration of VSMCs, thereby weakening the effect of smooth muscle cells on the progression of atherosclerosis, but the specific molecular mechanism is still unclear. Recently, Yu et al. [77] found that ATRA prevented vein grafts stenosis by inhibiting Rb-E2F-mediated cell cycle progression and KLF5-RARα interaction in human vein SMC. The mechanism of ATRA on smooth muscle cells should be further explored in atherosclerosis models in the future.

#### 4.2.2. ATRA and SMC Differentiation

There are two phenotypes of VSMCs: contractile VSMCs and synthetic VSMCs. This phenotypic differentiation occurs in response to microenvironmental stimuli. The phenotypic transition of SMCs plays a central role in the development, progression, and stabilization of plaques. “Phenotypic switching” is a process of medial SMC dedifferentiating, proliferating, and migrating to intimal lesions in response to atherosclerosis stimulation. Human genomics studies have found that the RA signal is one of the main regulators of this process. Activation of RA signaling by ATRA prevents the phenotypic transition of SMCs, reduces the burden of atherosclerosis, and promotes the stability of the fiber cap [78]. Retinoic acid is a well-known phenotypic differentiation inducer in many cell types, including vascular SMCs [79]. ATRA can enhance the expression of SMC differentiation markers smooth muscle MHC (myosin heavy chain) [80], α-actin [12,80,81,82,83,84], and significantly inhibit the protein expression and activity of matrix metalloproteinases MMP-2 and MMP-9 [80]. Colbert et al. [85] showed that a RARE-lacZ transgene colocalizes with the expression of adult smooth muscle myosin heavy chain subtype SM2. This is the first in vivo evidence supporting the role of activated retinoic acid receptors in SMC differentiation. Many in vitro studies have demonstrated that ATRA can affect the differentiation process of SMCs. In vitro, compared with SMCs in a normal culture medium, the sensitivity of intimal cells to ATRA-induced apoptosis is increased, confirming the main role of phenotype in determining SMC behavior [63,74,86]. Hayashi et al. [64] found that ATRA stimulated the expression of tropoelastin mRNA, which in turn promoted the fine expression of elastin in vascular SMCs of chicken embryos. Haller et al. [83] evaluated the effect of ATRA on the differentiation of SMCs in primary rat aorta by increasing the expression of protein kinase C(PKC)-α and SM α-actin. Fully differentiated SMCs can activate the myofilament device, increase intracellular calcium ions, and produce responsiveness to contractile agonists. Blank et al. [87] showed that the activation of G protein-coupled receptors cloned from ATRA-derived SMCs in P19 embryonal carcinoma cells (P19s) resulted in a significant increase in intracellular calcium. Parental P19s (untreated with ATRA) showed virtually no such increase in intracellular calcium levels. Wright et al. [88] showed that ATRA could restore the contractility of SMCs in aortic rings. In addition, Rogers et al. [89] found that ATRA increased the level of calcification inhibitor MGP (matrix Gla protein) and decreased the activity of TNAP (tissue-nonspecific alkaline phosphatase) through RARα and its coordinated transcriptional regulation, thereby inhibiting calcification of SMCs and aortic VICs.

Through the complex interaction with many factors, ATRA plays a pleiotropic effect on the differentiation, proliferation, and migration of VSMCs, and VSMCs play an increasingly important role in the pathogenesis of atherosclerosis. However, the sometimes contradictory nature found in the field highlights the need for further investigation of the mechanisms involved.

### 4.3. ATRA and Neutrophils

Neutrophils are the main effector cells of natural immunity, which fight infection through phagocytosis and degranulation. Similar to microglia/macrophages, neutrophils show heterogeneous phenotypes. N2 neutrophils promoted microglial/macrophage phagocytic activity and contributed to post-stroke inflammation reduction, whereas the N1 phenotype intensified neuroinflammation [90]. Activated neutrophils also release neutrophilic extracellular traps (NETs) in response to various stimuli [91]. The development of atherosclerosis, beginning with the formation of fatty streaks and progressing to atherosclerosis and plaque formation, relies on a chronic inflammatory process driven by lipids that involves blood vessels and immune cells [92]. It is known that hyperlipidemia-induced neutrophils have been shown to be positively correlated with human atherosclerosis and atherosclerosis-related diseases in humans [93,94]. Megens et al. [95] detected NETs in human plaques obtained by endarterectomy. Warnatsch et al. [96] found in a mouse model of atherosclerosis that sterile inflammation drives the production of cytokines that subsequently trigger neutrophils to release extracellular traps. A recent study shows that neutrophils are polarized toward beneficial N2 phenotypes with ATRA treatment, whereas the harmful NETs formation was inhibited [97]. In conclusion, ATRA regulates neuroinflammation after stroke and modulates neutrophilic function. Because of its protective effect against ischemic stroke, further clinical studies of ATRA as a promising treatment for patients with ischemic stroke are necessary.

### 4.4. ATRA and T Cell

A growing body of evidence supports the critical role of T cells as drivers and modulators of atherosclerosis, a chronic inflammatory disease. In atherosclerosis, T cells of different subtypes regulate the progression or regression of atherosclerosis by secreting inflammatory signals. Regulatory T (Treg) cells have an anti-atherosclerotic function, whereas CD4 + T helper 1 (Th1) cells and natural killer T cells have pro-atherosclerotic properties. Each Th cell subtype expresses unique characteristics and performs unique functions in the immune response process.

Th1 cells exert pro-inflammatory effects through binding to M1 macrophages and secreting IFNγ [98]; Th1 cells are the dominant T cell population in human plaques. A large number of studies have shown that Th1 cells have an atherosclerotic effect [99]. Since the Th2 phenotype is opposite to the differentiation of Th1 cells that promote atherosclerosis, Th2 cells seem to have anti-atherosclerotic effects. Cytokines produced by Th2 cells include IL-4, IL-5, and IL-13, all of which are considered to have atherosclerotic protection [100,101]. The influence of RA on the balance of the Th1/Th2 response is controversial. A large number of research studies have shown that ATRA can promote the differentiation of naive T cells into Th2 cells by inducing gene expression of IL-4 [102]. In addition, ATRA regulates the production of IL-12 through APCs, inhibits Th1 cell differentiation [103], and induces the expression of GATA3 (GATA-binding protein 3), an activator of transcription 6 (STAT6), which is important for maintaining Th2 response [102]. However, retinoic acid does not always inhibit Th1-related mechanisms. It is essential for the stability and maintenance of Th1 cells. RA signal transduction is essential for limiting the conversion of Th1 cells to Th17 effectors and preventing Th17 pathogenic reactions in the body [104].

TGFβ and IL-6 or IL-23 mediate differentiation into Th17 cells [105]. Th17 cells release the cytokines IL-17, IL-21, IL-22, and IL-23 [106]. Once Th17 cells infiltrate the plaque, SMCs are induced to secrete pro-inflammatory cytokines and chemokines [107]. The role of Th17 in atherosclerosis is still under debate. Treg cells are negative regulators of the immune system [108] and inhibit atherosclerosis by secreting anti-inflammatory cytokines including IL-10 and IL-35 [109]. The mechanism of RA’s influence on Th17/Treg balance is well known. In the presence of TGF-β [110], small intestinal lamina propria dendritic cells can synthesize RA and produce Treg. Therefore, the increase in TGF-β level promotes naive CD4 T cells to produce Treg through an RA-dependent mechanism [111,112,113,114].

Antigen-presenting cells can recognize pathogens directly. T cell activation requires antigen-presenting cells to recognize the antigen before proliferating and differentiating into effector cells [115]. Macrophages direct the responses of T cells, such as Th1, Th2, Th17, and Treg cells. Macrophages in M1 suppression mode direct T cells to produce Th1-like cytokines (such as IFN-γ), which stimulate specific cytolytic T cells and activate more M1 macrophages. In contrast, M2 macrophages stimulate T cells to produce Th2-like cytokines (such as IL-4 and TGF-β), causing B cell proliferation and antibody production, and further amplifying M2 responses [116]. It is known that most macrophages in progressive plaques in mice and humans resemble the activated classic M1 phenotypic state. Lin et al. found that plaque regression is characterized not only by the reduction in classically activated M1 macrophages but also by the enrichment of cells expressing selective activation (M2 or M[IL-4]) macrophage markers. It has been shown that activated M2 macrophages are involved in inflammation regression and tissue damage repair, which is consistent with the characteristics of plaque regression [117]. Macrophages can complete phenotypic transformation between M1 and M2 types in an ATRA-dependent manner, resulting in increased numbers of Tregs, which are also abundant in regression plaques. There has been no evidence to prove the direct regulation of ATRA on T cells in atherosclerosis, and further exploration is needed.

### 4.5. ATRA and Endotheliocyte

The formation mechanism of atherosclerosis is complex and diverse. The dysfunction and injury of vascular endothelial cells is one of the important risk factors for the development of atherosclerosis [118,119] and affects the degree of atherosclerotic lesions [120,121]. In the presence of a variety of risk factors for atherosclerosis, atherosclerotic lesions develop from local lesions to systemic complications, such as coronary heart disease and stroke [122]. Endothelial dysfunction is usually used to refer to the abnormal production or bioavailability of endothelial nitric oxide (NO), and the resulting harmful changes in vascular reactivity [123]. Studies have reported that ATRA may change the level of endogenous NO synthase inhibitor asymmetric dimethylarginine (ADMA), thereby increasing NO synthesis [124]. In addition, ATRA can inhibit the expression of endothelin-1 (ET-1) mRNA in endothelin-1 (ET-1) mRNA in endothelial cells (ECs), leading to anti-atherosclerosis effects [125,126]. ATRA may alleviate atherosclerosis by reducing ET-1 expression and caveolin-1 (CAV-1) expression in atherosclerotic rabbits and increasing endothelial nitric oxide synthase (eNOS) phosphorylation [127,128].

### 4.6. ATRA and Other Pathologic Mechanisms Associated with Atherosclerosis

ATRA can not only regulate the biological processes of immune cells, but also participate in the regulation of blood glucose concentration, oxidative stress, and other risk factors related to atherosclerosis [45,129,130]. Antidiabetic effects of ATRA in pancreatic beta cells have been reported [129,131], and high-dose vitamin A supplementation has been shown in a randomized trial to significantly reduce fasting blood glucose and glycated hemoglobin in patients with type 2 diabetes, but the results of this study failed to show a significant effect on lipid metabolism [132]. Furthermore, Blaner et al. [130] showed that ATRA mediates the antioxidant effects of vitamin A in vivo by affecting the transcription of genes critical for mediating antioxidant responses. They showed that the antioxidant effects of ATRA were dose-related, with low doses of ATRA having antioxidant properties, but high doses of ATRA increasing free radical production, inducing lipid peroxidation, and reducing cell viability.

## 5. Conclusions

The beneficial effects of ATRA and its isomers on cardiovascular disease outcomes have also been confirmed in many articles [75,120,133,134,135,136]. ATRA is currently used clinically to treat a variety of human malignancies [137,138,139], and its therapeutic dose, side effects, and interaction with other drugs have been clarified, which obviously has advantages in future clinical trials of atherosclerosis treatment. The current dose of ATRA in clinical use is 0.1–1 μM, and high doses of ATRA may bring serious adverse reactions [140]. With the development of nanotechnology, the targeted delivery of ATRA in the treatment of atherosclerosis has greater benefits and fewer adverse reactions [141]. The pathological process of atherosclerosis is complicated. An in-depth understanding of the effects of ATRA on the various stages of atherosclerosis may bring new progress in the treatment of atherosclerosis and new breakthroughs in the treatment of atherosclerotic cardiovascular diseases. Finally, the influence of other isomers of ATRA or synthetic retinoic acid on atherosclerosis cannot be ignored. In the future, more animal models are still needed to study the role of ATRA in atherosclerosis, especially in the immune response to various inflammatory cells. Further exploration is needed in the future.

## Figures and Tables

**Figure 1 biomolecules-12-00869-f001:**
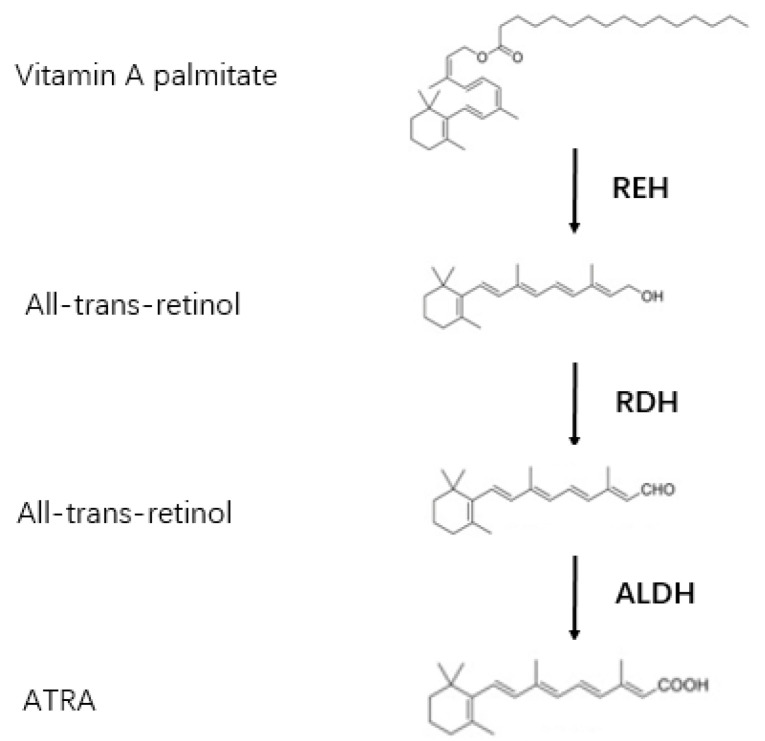
The process of converting of Vitamin A to ATRA.

**Figure 2 biomolecules-12-00869-f002:**
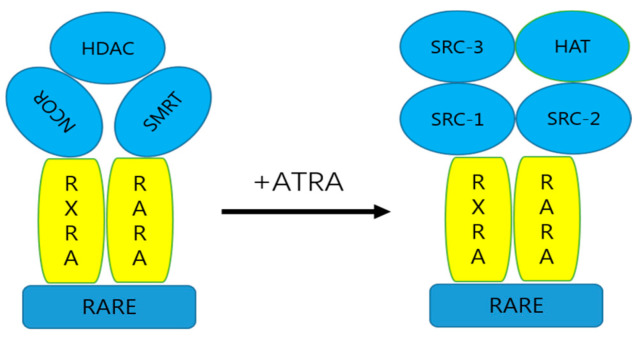
The genomic molecular pathway by ATRA. In the absence of ligands, retinoic acid X receptors and RXR/RAR heterodimers bind to RAREs in the regulatory regions of target genes and inhibit transcription by recruiting HDAC or silencing mediators (SMRT and NCoR). After binding to ATRA, the corepressor complex is dissociated, and the coactivator binds to histone acetyltransferase (HAT) activity, resulting in transcriptional activation of target genes.

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
