# Peer review of "Potential Therapeutic Effect of All-Trans Retinoic Acid on Atherosclerosis"

_biomolecules, 2022, doi:10.3390/biom12070869_

Round 1
Reviewer 1 Report
Comments:
Line 2: „acidon”
Line 14: „ In this paper, the effects of ATRA on lipid metabolism were reviewed” There is little information on ATRA-regulated lipid metabolism if any at all in, MS.
Line 36: suggested sentence: ATRA plays an important role in the cell growth, development, and differentiation of vertebrates.
Line 41 -43: „Studies have reported that ATRA can significantly inhibit the formation of atherosclerotic lesions in rabbit models of atherosclerosis induced by a high-fat diet. In addition, it was also shown that ATRA can inhibit restenosis after balloon.”
Atherosclerosis induced by a high-fat diet according to the reviewer may be an artifact. The rabbit is graminivorous!
Line 46: RA, abbreviation needs to be clarified at the first mention.
Line 74: „Transformation” is an incorrect word.
Line 72, 93: Figure 1 or 2 words should be inserted in full sentences, not separately.
Line 89: Full name of SRC is missing.
Line 85-86: Silencing mediator of retinoic acid and thyroid hormone receptor (SMRT), nuclear receptor corepressor (NCoR)
Line 97: Instead of NCORE be NcoR.
Line 116: WAT, abbreviation was explained before, use one of the two names, not both.
Line 117: apolipoprotein E-deficient mice (Apo-E).
Line 121: In vivo word should be in italic type.
Line 137. LDL, see Line 46
Line 143: Apo-E
Line 147: (HMDM), letter-spacing
Line 169: NLRP3, see Line 46
Line 189: VSNCs, see Line 46
Line 208-209: „uPA can activate a number of proteins, one of which is 208 PDGF-D, which is an effective activator of SMC proliferation.” Reference is missing.
Line 295: GATA3(GATA… ; transcription 6(STAT6) …. letter spacing
Line 324: „production of Tregs” production is an inappropriate word meaning „the action of making or manufacturing from components or raw materials, or the process of being so manufactured”
Line 334, 336, 339, and 341: , letter spacing
Line 344: „ATRA is currently used clinically to treat a variety of human malignancies,” References are missing.
Line 345: „rapeutic dose”
General remarks:
It should be pointed out or mentioned both when and on the basis of which biomarkers the treatment of ATRA should start during the formation of atherosclerotic plaque, as the process takes decades.
It should also be pointed out what ATRA concentration could be used for the treatment of atherosclerosis bearing in mind that ATRA toxicity exists.
Author Response
Dear Reviewer:
First of all, thank you very much for your time in reading and during revisions, and for your valuable comments. I have read your comments carefully and my paper with the following suggestions:
Amend it verbatim according to your comments;
Read carefully and revise the grammar and structure of the article;
Your "Approximately when ATRA occurs and on which biomarkers are based during most block formation when atherosclerotic problems begin treatment." By the beginning of the paper, after studies using ATRA, studies in atherosclerosis have been studied to aneurysmal plaques in massive atherosclerosis, and no biomarkers typical of atherosclerosis have been found the thing.
The ATRA used has been added to the summary section at the end of the main text. The usual therapeutic amount of ATRA is 0.1-1 μM.
this revision adds additional effects and anti-secondary effects of ATRA to the base.
Finally, thanks for your guidance, and thanks again for your review and hopefully corrections to my revised paper.
Sincerely,
Deng Qile

Reviewer 2 Report
The authors reviewed the role of All-trans retinoic acid (ATRA) in Atherosclerosis focusing on inflammation and other pathologic mechanism involved. in this pathology. The revision is complete and gives an idea about the field.
My suggestion is about the effect of ATRA on other pathologic mechanisms associated with atherosclerosis such as oxidative stress and glucose sensitivity.
Author Response
Dear reviewer:
First of all, thank you very much for taking the time out of your busy schedule to read and revise my paper and for your valuable suggestions. Regarding your proposed ATRA for other pathological processes associated with atherosclerosis has been added in section 4.6 of this paper.
Finally, I would like to express my thanks again for your guidance, and thank you for reviewing and correcting my revised paper again. I hope that with your guidance I can complete an excellent paper and sincerely hope that my paper will be published in your journal.
Sincerely,
Qile Deng
